# Reverse Genetics System for Shuni Virus, an Emerging Orthobunyavirus with Zoonotic Potential

**DOI:** 10.3390/v12040455

**Published:** 2020-04-17

**Authors:** Judith Oymans, Paul J. Wichgers Schreur, Sophie van Oort, Rianka Vloet, Marietjie Venter, Gorben P. Pijlman, Monique M. van Oers, Jeroen Kortekaas

**Affiliations:** 1Department of Virology, Wageningen Bioveterinary Research, Houtribweg 39, 8221 RA Lelystad, The Netherlands; judith.oymans@wur.nl (J.O.); paul.wichgersschreur@wur.nl (P.J.W.S.); sophie.vanoort@wur.nl (S.v.O.); rianka.vloet@wur.nl (R.V.); 2Laboratory of Virology, Wageningen University & Research, P.O. Box 16, 6700 AA Wageningen, The Netherlands; gorben.pijlman@wur.nl (G.P.P.); monique.vanoers@wur.nl (M.M.v.O.); 3Department Medical Virology, Faculty of Health Science, Centre for Viral Zoonoses, University of Pretoria, Pretoria 0028, South Africa; marietjie.venter@up.ac.za

**Keywords:** Shuni virus, Schmallenberg virus, orthobunyavirus, reverse genetics, reassortment

## Abstract

The genus *Orthobunyavirus* (family *Peribunyaviridae*, order *Bunyavirales*) comprises over 170 named mosquito- and midge-borne viruses, several of which cause severe disease in animals or humans. Their three-segmented genomes enable reassortment with related viruses, which may result in novel viruses with altered host or tissue tropism and virulence. One such reassortant, Schmallenberg virus (SBV), emerged in north-western Europe in 2011. Shuni virus (SHUV) is an orthobunyavirus related to SBV that is associated with neurological disease in horses in southern Africa and recently caused an outbreak manifesting with neurological disease and birth defects among ruminants in Israel. The zoonotic potential of SHUV was recently underscored by its association with neurological disease in humans. We here report a reverse genetics system for SHUV and provide first evidence that the non-structural (NSs) protein of SHUV functions as an antagonist of host innate immune responses. We furthermore report the rescue of a reassortant containing the L and S segments of SBV and the M segment of SHUV. This novel reverse genetics system can now be used to study SHUV virulence and tropism, and to elucidate the molecular mechanisms that drive reassortment events.

## 1. Introduction

The genus *Orthobunyavirus* (family *Peribunyaviridae*) is the largest genus within the order *Bunyavirales,* comprising over 170 named arthropod-borne (arbo) viruses divided over 18 serogroups [1,2]. The most widespread orthobunyavirus is Akabane virus (AKAV), a member of the Simbu serogroup, which is endemic to Africa, Australia, and Asia and causes congenital malformations in ruminants [3]. Other veterinary pathogens of this serogroup include Aino virus, Shamonda virus, and Peaton virus [4,5]. Apart from veterinary pathogens, the genus *Orthobunyavirus,* also comprises human pathogens, such as Oropouche virus (OROV), which causes a mild, self-limiting febrile illness [6,7]. Orthobunyaviruses causing more severe human disease include LaCrosse virus (LACV), the leading cause of pediatric arboviral encephalitis in the US, and Ngari virus, which was associated with large outbreaks of haemorrhagic fever in Africa [8,9].

Orthobunyaviruses contain a negative-strand RNA genome that is divided into three segments, named after their size, large (L), medium (M), and small (S) [2]. The L segment encodes the viral polymerase. The M segment encodes two structural glycoproteins, Gn and Gc, involved in receptor binding and the fusion of the viral and endosomal membranes. The M-segment additionally encodes a non-structural protein called NSm, which was suggested to function as a scaffold for virion assembly [10]. The S segment encodes the nucleocapsid protein (N) and a non-structural protein (NSs) in overlapping open reading frames. The NSs protein is considered the major virulence determinant of orthobuyaviruses by antagonising host innate immune responses, including type I interferon responses [11,12]. 

On November 2011, a previously unidentified orthobunyavirus was detected in the blood of a diseased cow from a farm near the German town Schmallenberg. The so called Schmallenberg virus (SBV) belongs to the Simbu serogroup and was first associated with fever, diarrhea, and reduced milk yield, but was later found to be the causative agent of severe congenital malformations in ruminants, manifesting with arthrogryposis and hydranencephaly [13,14]. SBV was shown to be transmitted by *Culicoides* biting midges and spread rapidly across north-western Europe, ultimately being reported in 27 countries by September 2013 [15]. Although the number of cases of congenital malformations have decreased since the outbreak, SBV is still circulating in Europe. Analysis of serum samples collected from over 300 people who lived at or near a farm where SBV was found revealed no SBV-specific antibodies, suggesting that SBV is not infectious to humans [16]. 

Shuni virus (SHUV) is another member of the Simbu serogroup, which was first isolated in the 1960s in Nigeria from a slaughtered cow and from the blood of a febrile child [17]. In 1977, SHUV was isolated from the brains of two horses that had succumbed to neurological disease, one in South Africa and one in Zimbabwe [18,19]. The virus re-emerged in 2009 in South Africa, where it was again associated with the neurological disease of horses [20]. Analysis of serum samples collected from veterinarians revealed a seroprevalence of 4%, suggesting that SHUV, in contrast to SBV, is infectious to humans [21]. In 2014, SHUV emerged in Israel, where it was associated with congenital malformations in sheep, goat and cattle and fatal neurological disease in young cattle [22,23,24]. SHUV has been isolated from both field-collected midges and *Culex theileri* mosquitoes [25,26]. Recently, the virus was shown to infect and to disseminate in two laboratory-reared *Culicoides* species: *C. nubeculosus* and *C. sonorensis* [27]. In the same study, SHUV did not disseminate to the saliva of two mosquito species, *Culex pipiens pipiens* and *Aedes aegypti*. 

An important feature of orthobunyaviruses is their ability to reassort their genome segments with related viruses [28,29]. The exchange of the M segment may lead to changes in host tropism as the glycoproteins mediate cell entry. Whereas SBV was considered a novel orthobunyavirus upon discovery, careful phylogenetic analysis has suggested that SBV is an ancestor of Shamonda virus (SHAV), which appears to contain the L and S segments of SBV, while the M segment was obtained from another, unidentified orthobunyavirus [30]. Despite belonging to the same serogroup, there is generally little cross neutralization between members of the Simbu serogroup [31]. Therefore, M segment reassortants may be capable of replicating in animals or humans that were infected previously by another member of the same serogroup. What drives the reassortment events and under which conditions segment exchange takes place is largely unknown. Of note, accumulating evidence suggests that compatibility of the viral RNA polymerase (encoded by the L-segment) and the nucleocapsid protein (encoded by the S-segment) is an important determinant of reassortant potential [32].

We here report the development of a reverse genetics system for SHUV that can be used to study virulence and tropism. The system was validated by creating a SHUV mutant lacking NSs expression and was subsequently used to evaluate viability of reassortants of SBV and SHUV, resulting in the rescue of a novel orthobunyavirus containing the S and L segment of SBV with the M segment of SHUV. 

## 2. Materials and Methods

### 2.1. Cells and Cell Culture

Culture media and supplements were obtained from Gibco (Thermo Fischer Scientific, Breda, The Netherlands) unless indicated otherwise. BSR cells constitutively expressing T7 RNA polymerase (BSR-T7 cells) were kindly provided by Prof. Klaus Conzelmann (Ludwich-Maximilians-Universität, München). These cells were maintained in Glasgow minimum essential medium (GMEM) supplemented with 1% minimum essential medium nonessential amino acids (MEM NEAA), 4% tryptose phosphate broth, 1% antibiotic/antimycotic (a/a), and 5% foetal bovine serum (FBS) (complete medium). Vero E6 cells were maintained in minimum essential medium (MEM) supplemented with 1% a/a, 5% FBS, 1% glutamine and 1% MEM NEAA. Human A549 cells were cultured in Dulbecco’s modified Eagle medium (DMEM) supplemented with 10% FBS and 1% a/a. A549, Vero E6 and BSR-T7 cells were cultured at 37 °C with 5% CO_2_. 

An immortalized sheep liver cell line was developed by isolating hepatocytes from the liver of a 4-week-old lamb. The liver was rinsed by injecting Hank’s Balanced Salt Solution (HBBS) in the portal vein. Next, the liver was cut into small slices that were incubated with 0.1% collagenase IV in HBBS for 30 min at 37 °C. Liver cells were filtered through a 70-µm cell strainer and subsequently centrifuged 5 min at 300× *g*. The pellet was resuspended in Ammonium-Chloride-Potassium (ACK) lysis buffer (Thermo Fisher Scientific, Breda, The Netherlands) and incubated for 5 min at room temperature. The cells were washed twice with HBBS and cultured in collagen-coated T75 flasks. The cells were maintained in DMEM supplemented with 10% FBS, 0.04% supplement B, 0.02% epidermal growth factor (EGF), and 1% a/a at 37 °C with 5% CO_2_. After 2 cell passages, the cell cultures were sent to InSCREENeX GmbH, Braunschweig, Germany, to immortalize the cells as described by Lipps et al. [33]. The cells were maintained in collagen (InSCREENex)-coated flasks with DMEM supplemented with 10% FBS and 1% a/a at 37 °C with 5% CO_2_. The immortalized cell line was named Ovine Hepatocyte Cell #3 (OHC3).

Sf9ET cells (ATCC^®^ CRL-3357™) were cultured in Insect-XPRESS medium (Lonza, Maastricht, The Netherlands) supplemented with 1% a/a. High Five cells were maintained in Express Five medium supplemented with 1% glutamine and 1% a/a. Both these insects’ cell lines were cultured in suspension at 28 °C in a shaking incubator. 

### 2.2. Rescue of SBV and SHUV from cDNA

The full-length cDNA of the L, M, and S genome segments from SBV strain BH80/11-4 (GenBank numbers: HE649912.1, HE649913.1 and HE649914.1) and SHUV strain 18/09 (GenBank numbers: NC_043699.1, NC_043698.1, NC_043697.1) were synthesized and cloned in antigenomic-sense orientation in pUC57 vectors, flanked by a T7 promoter at the 5′ end and a hepatitis δ ribozyme and T7 terminator at the 3′ end by the GenScript Corporation (Piscataway, NJ, USA), generating pUC57-SBV-S, pUC57-SBV-M, pUC57-SBV-L, pUC57-SHUV-S, pUC57-SHUV-M, and pUC57-SHUV-L. To determine the nucleotide and amino acid identity between the SBV and SHUV segments, the NCBI BLAST sequence analysis tool was used [34]. To create SBV and SHUV NSs deletion mutants, 4 stop codons were introduced downstream of the methionine codons in the NSs genes without changing the amino acid sequences of the N proteins generating pUC57-SBV-S-∆NSs and pUC57-SHUV-S-∆NSs. The first three mutations in SBV∆NSs are as described by Elliott et al. [35]; however, a fourth stop codon was introduced to prevent translation from a fourth methionine further downstream. 

To optimize rescue efficiency, BSR-T7 cells were infected prior to transfection with a recombinant fowlpox virus expressing T7 polymerase (fpEFLT7pol), here referred to as FP-T7, kindly provided by the Institute for Animal Health (IAH, Compton, UK). Specifically, BSR-T7 cells were seeded in T75 flasks in complete medium with 10% Opti-MEM (Gibco, Thermo Fischer Scientific, Breda, The Netherlands) 1 day before transfection. The medium was replaced by 5 mL Opti-MEM containing 10^6.5^ TCID_50_ of FP-T7 and incubated at 37 °C and 5% CO_2_ for 2 h. The medium was subsequently removed, and the cells were washed once in Opti-MEM after which 5 mL fresh Opti-MEM was added. The transfection mixture was prepared according to manufacturer’s instructions (*Trans*IT^®^-LT1; Mirus Bio, Madison, WI, USA). Briefly, highly pure 7.5 µg pUC57-L, 7.5 µg pUC57-M and 5 µg pUC57-S (of SBV or SHUV) were added to NaCl, before adding LT1 transfection reagent according to manufacturer’s protocol. With the aim to rescue reassortants, plasmids encoding SHUV L, M, or S segments were replaced for the corresponding plasmids of SBV. All 6 combinations were transfected and subsequently incubated at 37 °C and 5% CO_2_. Four hours post transfection, 10 mL complete medium was added. At 5 days post transfection, supernatants were harvested and passaged on Vero-E6 cells in 6-well plates. Larger stocks were prepared by inoculating T150 flasks containing 6 × 10^6^ Vero E6 cells at a multiplicity of infection (MOI) of 0.01. The virus was harvested at four days post infection. Each rescue experiment with the aim to rescue reassortant viruses was performed three times.

To confirm the identities of the wild type, ∆NSs mutant viruses and reassortants, reverse transcriptase (RT)-PCR and Sanger sequencing were performed. To this end, 200 µL virus stock solution was added to 2 mL EasyMAG lysis buffer (Biomérieux, Amersfoort, The Netherlands), followed by RNA isolation by the NUCLISENS^®^ EasyMAG robot (Biomérieux, Amersfoort, The Netherlands). Next, cDNA was generated using the Superscript IV kit (Thermo Fisher Scientific, Breda, The Netherlands) with random hexamers according to manufacturer’s protocol. The NSs coding region of the S-segment was amplified by PCR, using the Phusion High-Fidelity PCR Master mix (Thermo Fisher Scientific). For SHUV, a 320-bp fragment and for SBV a 545-bp fragment was amplified using primer sets 1 and 2, respectively, as described in Table 1. The PCR products were concentrated and purified with the PCR clean-up kit (Macherey-Nagel, Bottrop, Germany) according to manufacturer’s protocol. Purified fragments were sent to Macrogen (Amsterdam, The Netherlands) for sequencing. 

To confirm the identity of the recombinant wild type and reassortant viruses, segment- and virus-specific reverse transcriptase PCR (RT-PCR) reactions were performed using the primers listed in Table 1 (primer sets 3–8). The primers were ordered at Integrated DNA technologies, Leuven, Belgium. The different DNA fragments were visualised with an Agilent 2200 Tapestation system in combination with a D1000 Screentape (Agilent, Amstelveen, The Netherlands). 

### 2.3. Production of Rabbit Antisera Against the SBV and SHUV Gc Head Domains (Gc_head_)

The N-terminal 234 amino acids of the Gc protein of SBV, here referred to as Gc_head_, were previously shown to be highly immunogenic [36]. With the aim of developing rabbit antisera against the Gc protein of SBV and SHUV, we developed constructs encoding the Gc_head_ of both viruses. DNA fragments encoding the N-terminal 234 amino acids of the SBV Gc protein or 255 amino acids of the SHUV Gc protein flanked by a N-terminal GP64 signal sequence and a C-terminal twin Strep-Tag were synthesised by GenScript (Piscataway, NJ, USA). Following the cloning of the fragment in a pBAC-3 baculovirus vector, recombinant baculoviruses were generated with the flashBAC™ ULTRA baculovirus expression system (Oxford Expression Technologies, Oxford, UK). Briefly, transfection mixtures containing the pBAC-3 vector, a bacmid, and cellfectin II were added to 24-wells plates containing 200,000 SF9ET cells per well. Rescued viruses were subsequently amplified in SF9-ET suspension cultures infected at low MOI. For protein production, High Five cells were infected at high MOI according to the manufacturer’s protocol (Thermo Fisher Scientific, Breda, The Netherlands). Proteins were purified from supernatants using Strep-Tactin^®^ resin (IBA, Göttingen, Germany) according to the manufacturer’s instructions. Buffers were exchanged to Tris-buffered saline + 200 mM NaCl using Amicon^®^ Ultra centrigufal filters (Merck-Millipore, Amsterdam, The Netherlands). Proteins were separated in 4–12% SDS gels (Thermo Fisher Scientific, Breda, The Netherlands) and stained with GelCode Blue Stain reagent (Thermo Fisher Scientific, Rockford, USA). Two New Zealand White rabbits per viral protein were subsequently immunised with 1 mg of protein by Genscript (Piscataway, NJ, USA) following their standard polyclonal antibody service protocols for the production of rabbit antisera. 

### 2.4. Immunofluorescence Assays

Vero E6 cells (25,000/well) were seeded in Grace Bio-Labs CultureWell™ removable chambered cover glasses (Sigma-Aldrich, Zwijndrecht, The Netherlands) in complete medium. The cells were subsequently infected with virus at an MOI of 1. After 24 h, the cells were fixed in 4% paraformaldehyde for 30 min and permeabilized using a 1% Triton-X-100 in PBS solution. To stain the infected cells, the primary polyclonal rabbit sera against SBV-Gc_head_ or SHUV-Gc_head_ (in PBS supplemented with 5% horse serum, Gibco, Thermofisher Scientific, Breda, The Netherlands; diluted 1:4000, 2 h at 37 °C) in combination with a donkey anti-rabbit IgG Alexfluor568 (Life Technologies, Thermofisher Scientific, Breda, The Netherlands, in PBS supplemented with 5% horse serum, Gibco, Thermofisher Scientific, Breda, The Netherlands; diluted 1:500, 1 h at 37 °C) secondary antibody was used. The nuclei were visualized using 4′,6-diamidino-2-phenylindole (DAPI) and cells were submerged in Vectashield (H-1000, Vector Laboratories, Peterborough, UK) prior to imaging. The images were obtained with an inverted fluorescence wide-field ZEISS Axioskop 40 microscope with appropriate filters and a 1.3 NA 100× oil objective in combination with an Axiocam MRm CCD camera.

### 2.5. Virus Neutralization Test

Sera were heat-inactivated at 56 °C for 30 min and serially diluted (1:3) in 96-well plates with a starting dilution of 20. Subsequently, 50 µL of virus was added with a titre of 10^4.7^ TCID_50_/mL, after which the samples were incubated at RT for 2 h. Next, 30,000 Vero cells in 100 µL were added to each well. The plates were incubated for 48 h at 37 °C with 5% CO_2_. All dilutions and suspensions were prepared in complete medium. After 48 h, an immunoperoxidase monolayer assay (IPMA) was performed as described previously using the rabbit antisera as primary antibodies and anti-goat-HRP (Dako, Agilent, Santa Clara, CA, USA) as a secondary antibody [37]. A well was scored positive for neutralization if less than 50% of the cells were stained, the negative control (without serum) was used as 100% cell staining. All samples were tested in triplicate. Titres were defined as the average of the highest dilution that was positive for (>50%) neutralization. 

### 2.6. Growth Curves

A549 and Vero cells (750,000) were seeded in T25 flasks and infected with virus the following day at an MOI of 0.01 in 5 mL complete medium. Directly upon infection, a 250-µL sample was taken and stored at −80 °C until analysis. At 2 h post infection, the cells were washed and the medium was replaced and another sample was collected. At 24, 48, and 72 hpi, additional samples were collected. All growth curves were performed in triplicate and each sample was titrated in triplicate using end-point dilution assay as described previously [38]. An immunoperoxidase monolayer assay (IPMA) was performed as described using the rabbit antisera as primary antibodies and anti-rabbit-HRP (Dako, Agilent, Santa Clara, CA, USA) as a secondary antibody [37]. Titres were determined using the Spearman-Kärber algorithm and were expressed as TCID_50_/mL. 

## 3. Results

### 3.1. Rescue of Wild-Type rSHUV and rSHUV-ΔNSs

To establish a reverse genetics system for SHUV, plasmids were designed to encode full-length S, M, and L segments in antigenomic-sense orientation, flanked by a T7 promoter at the 5′ ends and a T7 terminator and hepatitis delta virus ribozyme at the 3′ ends. The DNA constructs were synthesized and cloned into pUC57 plasmids by the GenScript Corporation, resulting in pUC57-SHUV-S, pUC57-SHUV-M, and pUC57-SHUV-L (Figure 1A). As a control, plasmids encoding SBV antigenomic-sense full-length S, M, and L were constructed. The nucleotide and amino-acid identities between SBV and SHUV are indicated in Figure 1B. For each virus, S plasmids with abrogated NSs genes were also constructed. Considering that the NSs and N proteins are encoded by overlapping open reading frames, we introduced four stop codons into the NSs genes without changing the amino acid sequences of the N proteins. Each stop codon was introduced downstream of a methionine, to prevent the possible production of truncated NSs proteins (Figure 1C). Following transfection, cytopathic effects (CPEs) were observed and supernatants were harvested and transferred to Vero cells to produce virus stocks (Figure 1A). Virus identities were subsequently confirmed by RT-PCR and the Sanger sequencing of the S segment amplicons. 

### 3.2. Replication of rSHUV and rSHUV∆NSs in Interferon-Competent and Interferon-Incompetent Cells 

Previous studies have shown that SBV∆NSs replicates poorly in IFN-competent cells, whereas wild-type SBV and SBV-∆NSs replicate equally efficiently in cells deficient in type I IFN responses [11,12]. The 80% amino acid identity of SBV and SHUV NSs (Figure 1B) suggests that the deletion of SHUV NSs would result in a similar phenotype. To test this hypothesis, the growth kinetics of SBV, SBV-∆NSs, SHUV, and SHUV-∆NSs were compared in different cell lines. In agreement with the aforementioned studies, the deletion of NSs expression had no influence on replication in Vero cells in which both wild-type and SBV∆NSs replicated to 10log 6 TCID_50_/mL (Figure 1D). Growth was subsequently evaluated in IFN-competent cell lines, A549 cells, and an immortalized ovine hepatocyte cell line (OHC3), which was developed in the present work. Replication of SBV∆NSs was impaired in A549 cells as the final titre of SBV∆NSs was 3 logs lower than the wild type (Figure 1E), and, also, in OHC3 cells, a final difference of 2 logs was noted (Figure 1F). Next, the growth kinetics of SHUV and SHUV∆NSs were compared in the different cell lines. No difference in the replication in Vero cells were observed (Figure 1G). However, a 3-log difference in growth was observed in A549 and OHC-3 cells, although the attenuation of SHUV∆NSs was less clear in A549 cells due to a large variation in growth (Figure 1H,I). 

### 3.3. Rescue of an SBV/SHUV Reassortant

The ability of orthobunyaviruses to reassort their genome segments was previously shown to be mediated by the panhandle sequences in the UTRs [32]. The alignment of the genome segment UTRs revealed high identity between the SBV and SHUV panhandles of each segment, with the exception of position 8 in the M and S UTRs (Figure 2A). We were successful in rescuing a virus containing the L and S segment of SBV, and the M segment of SHUV. However, we were unable to rescue all other possible reassortants, including the vice versa—a virus containing the S and L segments of SHUV and the M segment of SBV. The identity of the reassortant virus (rSBV_LS_/SHUV_M_) was confirmed by conventional PCR (Figure 2B). To further confirm the identity of the reassortant, an immunofluorescence assay was performed using antisera that specifically recognize SBV or SHUV Gc_head_. Antisera raised against SBV Gc_head_ only recognized SBV, whereas antisera raised against SHUV Gc_head_ specifically recognized SHUV as well as the reassortant, but not SBV (Figure 2C).

The successful rescue of a virus containing SBV L and S and SHUV M, suggests that such a reassortant could also emerge in the field. It was therefore interesting to investigate if this virus could be neutralized by SBV or SHUV-convalescent sera. Whereas SHUV convalescent serum was not available, SBV convalescent serum from sheep neutralized SBV, but not SHUV or the reassortant. In agreement with this result, the SBV-Gc_head_ antiserum neutralized SBV only, whereas the SHUV-Gc_head_ antiserum neutralized both SHUV and the reassortant (Figure 2D).

Finally, we compared the growth dynamics of the SBV_LS_/SHUV_M_ with wild-type SHUV and SBV in IFN-competent and IFN-incompetent cell lines. Interestingly, SBV_LS_/SHUV_M_ replicated equally efficiently as wild-type SHUV and SBV in Vero cells (Figure 2E). However, it was striking to find that the reassortant replicates poorly in the IFN competent cells A549 and OHC3 (Figure 2F,G).

## 4. Discussion

We report a reverse genetics system that can be used to study SHUV virulence and tropism. Recombinant SHUV was shown to replicate more efficiently than rSBV in Vero, A549 and OHC3 cells. In agreement with studies previously performed with SBV lacking NSs, both rSBV∆NSs and rSHUV∆NSs were significantly attenuated in IFN-competent cells, attributed to the IFN antagonistic function of orthobunyavirus NSs proteins [11,12].

The emergence of SHUV in Africa and Israel, its association with both congenital malformations and fatal encephalitis in ruminants [22], horses and wildlife [20,39], and its recent association with neurological disease in humans [40], calls for awareness and the development of control tools. Furthermore, areas where related orthobunyaviruses are endemic should also anticipate the emergence of reassortants. Considering that SBV is endemic to Europe, we used our SHUV and SBV reverse genetics systems to evaluate the viability of reassortants of these two viruses. All six combinations of genome segments were evaluated, whereas only the combination of SBV S and L with the M segment of SHUV was shown to be viable. These findings are strikingly similar to those of a previous study in which the possibility of SBV and OROV to reassort was evaluated, using minigenome-reporters and virus-like particles (36). Specifically, this previous study demonstrated that SBV N and L proteins can use the promoter of the M segment of OROV, whereas the vice versa did not result in genome replication. From this, it was concluded that only one viable virus may result from the reassortment of SBV and OROV, a virus containing the L and S segment of SBV and the M segment of OROV. The authors of this previous study speculated that two residues in the panhandle of the M segment, at positions 8 and 9, play a role in determining the ability to reassort. The SHUV UTRs, like the UTRs of OROV, Bunyamwera virus, Oya virus, and Perdoes virus, contain one mismatch at position 9, while the SBV UTR contains a double mismatch, which is also the case for Akabane virus (AKAV). These positions have been shown to be important for promotor activity [32,41]. Particularly position 8 of the 5′ UTR forms highly specific protein-RNA hydrogen bonds. A difference in the promotor binding site of SBV and AKAV may enable the recognition of another nucleotide. This suggests that the polymerases of AKAV and SBV are more promiscuous to accept/recognize the genome segments of other bunyaviruses and thus are more likely to reassort. The same mismatch is found in the UTR of the S segment; however, as it seems to be the case that the L and S segments of the genome have to be from the same virus, this mismatch is probably redundant for the possible formation of a viable reassortant. This previous work and our present study underscore that further research is needed to understand the molecular basis of this apparently restricted reassortment potential.

It was striking to observe that the SBV/SHUV reassortant replicated more efficiently in Vero cells than SBV and equally efficiently as SHUV. In strong contrast, the replication of the reassortant was reduced in the IFN competent A549 and OHC3 cells. This phenotype is possibly related to the suboptimal compatibility of the SBV and SHUV genome segments and/or proteins, which may be improved upon by further passaging of the reassortant virus. The interaction between the SHUV M segment and SBV N protein could be suboptimal, resulting in a larger amount of unbound double-stranded RNA in the cytosol, which could form secondary RNA structures, normally not found in bunyaviruses, and stimulate interactions with cytoplasmic sensors, such as MDA-5 [42].

Considering the limited cross-neutralization between members of the Simbu serogroup, it was plausible to assume that the M segment reassortant is not neutralized by SBV-specific antibodies [31]. Indeed, the reassortant was not neutralized by a sheep convalescent antiserum against SBV. As a convalescent serum with SHUV-neutralizing activity was not available, rabbits were immunized with baculovirus-produced Gc_head_. The corresponding protein from SBV was previously shown to induce high levels of neutralizing antibodies, which was taken along in this experiment as a positive control for SBV neutralization and as a negative control for SHUV neutralization. The antiserum against SHUV Gc_head_ effectively neutralized rSHUV and the reassortant, whereas the cross-neutralization of both sera was not observed. This finding confirms that the Gc_head_ domain is a very powerful immunogen with potential use for vaccine applications, with the important note that protection may be species specific.

In conclusion, the SHUV reverse genetics system developed in the present work can be used to further study the interactions of SHUV with its vertebrate and insect hosts. It would additionally be interesting to evaluate if SHUV can form viable reassortants with other orthobunyaviruses and to study the virulence and tropism of these novel viruses.

## Figures and Tables

**Figure 1 viruses-12-00455-f001:**
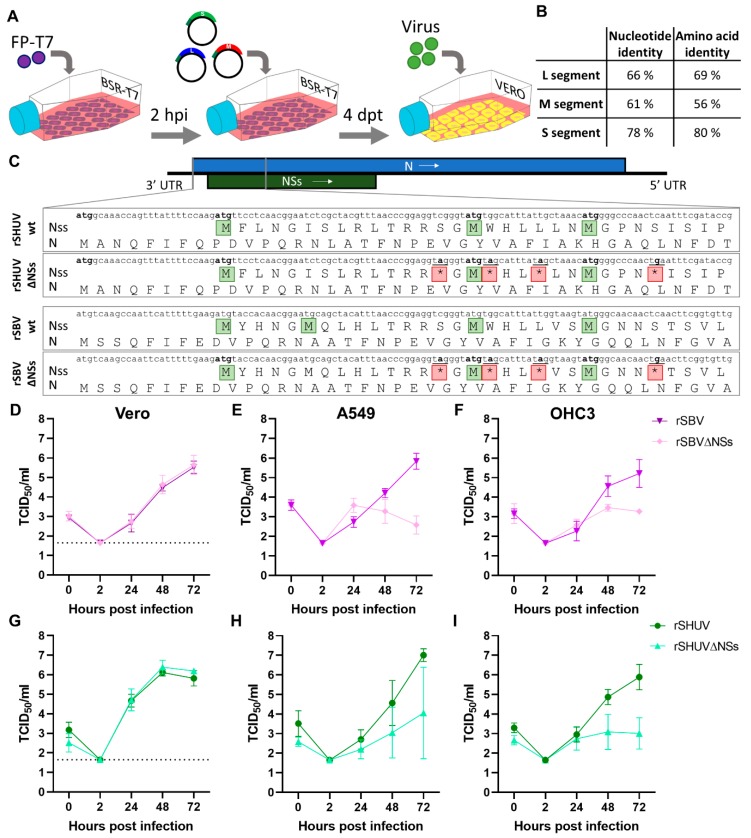
Rescue and growth characterization of recombinant Shuni virus (SHUV). (**A**) A schematic presentation of the reverse genetics system. Briefly, BSR-T7 cells were incubated with FP-T7. Two hours later, the medium was replaced, and the cells were transfected with pUC57 plasmids encoding each of the three genome segments. Supernatants containing recombinant virus were harvested at 4 days post transfection and transferred to Vero cells. The virus was harvested from the Vero cells at 4 dpi. (**B**) Nucleotide and amino acid identity between wild-type rSBV and rSHUV. (**C**) Partial sequences of the S-segments of rSHUV, rSHUV∆NSs, rSBV, and rSBV∆NSs. The start codons are marked by green boxes. The four introduced stop codons are indicated by red boxes. The replication of rSBV (purple) and rSBV∆NSs (pink) in Vero cells (**D**), A549 (**E**), and OHC3 cells (**F**). The replication of rSHUV (light green) and rSHUV∆NSs (dark green) in Vero (**G**), A549 (**H**), and OHC3 cells (**I**). The cells were infected at a multiplicity of infection (MOI) of 0.01, and the samples were collected at 0, 2, 24, 48, and 72 hpi. At 2 hpi, the inocula were removed, the cells were washed, and fresh medium was added.

**Figure 2 viruses-12-00455-f002:**
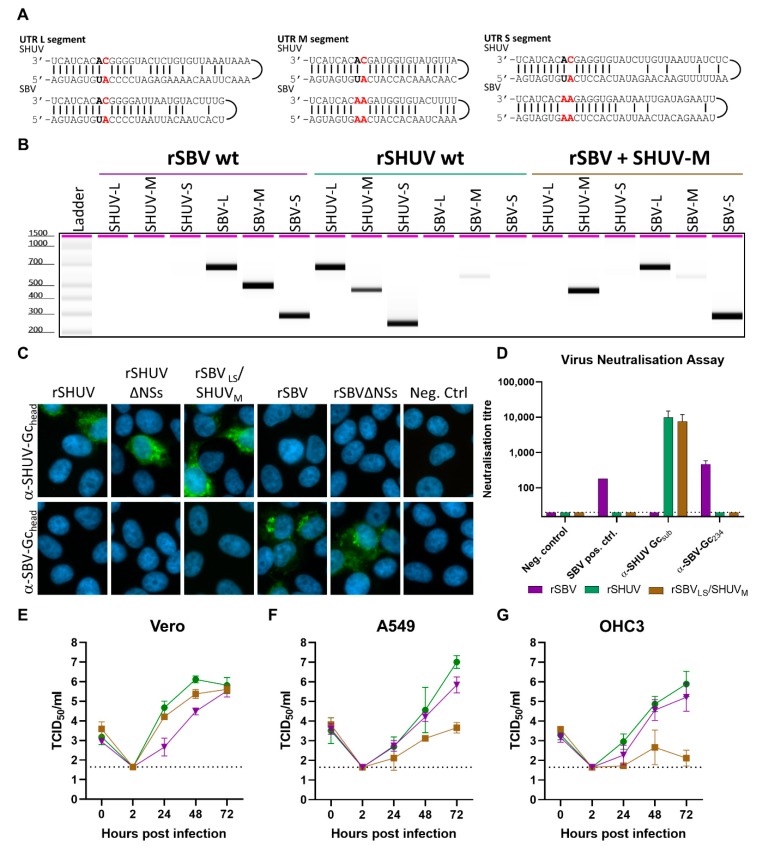
Characterization of a Schmallenberg virus (SBV)-SHUV reassortant. (**A**) The alignment of the UTRs of SBV and SHUV. SHUV contains a single mismatch at position 9, while SBV contains a double mismatch at positions 8 and 9 (indicated in red) in the UTR of the M and S segment. (**B**) The identity of the reassortant was confirmed by PCR using primers specifically recognizing SBV or SHUV L-, M-, or S-segments. PCR products were analyzed with the Tapestation system. (**C**) The immunofluorescence staining of Vero cells infected with rSHUV, rSHUV∆NSs, rSBV, or rSBV∆NSs at 24 hpi. The cells were infected using an MOI of 0.01. (**D**) Virus neutralization assay with a convalescent sheep serum against SBV, and rabbit antisera raised against the Gc head domains of SBV or SHUV. Replication of rSBV (purple), rSHUV (green) and the reassortant (rSBV_LS_/SHUV_M_), brown) in Vero (**E**), A549 (**F**) and OHC3 cells (**G**). The cells were infected at an MOI of 0.01, and the samples were collected at 0, 2, 24, 48, and 72 hpi. After 2 h, the medium was removed, the cells were washed, and fresh medium was added.

**Table 1 viruses-12-00455-t001:** Primers used in conventional PCR.

Primer Set	What	Primer Sequence (′5-′3)	Length Expected Product
1	S segment SHUV	ATGGCAAACCAGTTTATTTTCCA	320 bp
		TGATCTGCAACCCATTTTGC	
2	S segment SBV	GTGAACTCCACTATTAACTACAG	545 bp
		TCCATATTGTCCTTGAGGACCCTATGCATT	
3	L segment SHUV	AGAGAAAACAATTCAAAATGGATCCTTACC	654 bp
		TAAGTGAGTTGTAAAACTCTTTGAATATAGGATGAGTA	
4	M segment SHUV	TGGAGAGCTGGTGAAAACTGTCA	432 bp
		GTTTTGAGGCCACAAGTGACATC	
5	S segment SHUV	AGAACAAGTTTTTAAATGGCAAACCAGT	238 bp
		TAACCAATGTAAATTTGATGCCACCAAATG	
6	L segment SBV	CATGGCTAGACATGACTACTTTGGTAG	666 bp
		AAAATGTTATAATCATTGCCATATCTATTTATAACCTTTTGT	
7	M segment SBV	CCTGTTTAGCTTTTGCACTCCC	483 bp
		CACATGTTACCTCAATGGATTCGC	
8	S segment SBV	TTGAAGATGTACCACAACGGAATGCA	286 bp
		CGTGCTAGATATCCTGACATCCTG	

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
