# Peer review of "Reverse Genetics System for Shuni Virus, an Emerging Orthobunyavirus with Zoonotic Potential"

_viruses, 2020, doi:10.3390/v12040455_

Round 1
Reviewer 1 Report
In the manuscript authors describe reverse-genetics system for reassortant SBV and SHUV. I recommend to work on structure of the manuscript all over the text.
Lines 244-260. All paragraph was to be rewritten in shorter sentences. Not clear. It should include only result without discussing the results.
All over the text not only manufacturers should be shown, but also their addresses.
Abstract.
Lines 23-24. “The zoonotic potential of SHUV was recently underscored by its association with
neurological disease in humans”. It is controversial. ” I recommend to transfer the sentence into introduction or even discussion and to change the word “underscored to “unknown” or “not clear”. I suppose authors mean “febrile symptoms and neurological disease in overlooked patients in Gauteng hospitals has led to the identification of the several cases of the unrelated West Nile virus infection (Zaayman, 2012)”. If so, all these facts should be better described.
Line 27. “tropism” . Not clear what authors mean. Tropism for different hosts of tissues, or both.
Introduction.
Description of “R&D Blueprint” and consecutive description of order Bunyavirales (family, genus, serogroup) should be written separately. I recommend to pay more attention to viruses belonged to Simbu serogroup viruses, than to other viruses belonged to order Bunyavirales.
Probably, authors wanted to show SHUV as a potential human pathogen. If so, the authors should be consequent showing this data and writing their supposition.
Line 33-34. Explanation for “R&D Blueprint “has to be added.
Line 42-43. “The most widespread orthobunyavirus, named Oropouche virus (OROV), a member of the Simbu serogroup, causes a mild, self-limiting febrile illness [2, 3]”. Species OROV orthobunyavirus, which includes several different viruses, spreads in South America only, which is really big), causing disease in human. Describing OROV in such context is not related to the issue unless authors want to provide an example of simbuvirus causing illness in human. The most widespread virus is Akabane virus, which present in Africa, Middle East, Asia and Australia. Please, make changes.
Line 59. After the sentence “In 1977, SHUV was isolated from the brains of two horses that had succumbed to neurological disease.” reference has to be added. Please, add information in which geographic area SHUV was isolated in 1977.
Line 63. I suggest to change slightly the text . “where it was associated with congenital malformations” to add “in domestic ruminants”
Line 64. Neurological sings and fatalities were observed in young cattle only. Please, make changes.
Lines 84-85. “ Therefore, M segment reassortants may be capable of replicating in animals or humans that were infected previously by another member of the same serogroup [25]. “ Reference used in the text based on 24 Sumbu serogroup members serological analyses. The data is incomplete. Now 32 members are already known. Please, rewrite the text.
Line 111. “ACK (Ammonium-Chloride-Potassium)”- abbreviation and full name of the lysis buffer should exchange their places.
Line 113. RT, please first write full name and afterword in brackets abbreviation.
Line 120. Please, provide an official name of High Five cells. “Express Five medium”. Please, add manufacturer of medium.
Line 148. Please, add temperature of incubation.
Line 152. RT-PCR- add full meaning of the abbreviation.
Lines 185-188. Number of permit for using animals should be added.
Line 198. Full name of DAPI should be added
Line 222. (DAKO). Not only manufacturer should be shown, but also its address.
Results.
Lines 227-230. “The most widely employed method to rescue (ortho)bunyaviruses from cDNA makes use of three plasmids encoding antigenomic-sense RNA of each of the three genome segments, flanked by
a T7 promoter at the 5’ end and a T7 terminator and hepatitis delta virus ribozyme at the 3’ end (Fig. 1A) [29, 33, 34]. “ This sentence is not related to results.
Line 230. “These three plasmids are transfected into cells expressing T7 polymerase.” Names of plasmids have to be written. The word ”Here” has no need.
Line 257. “Similar results were obtained” has no need to be written.
Lines-276-279.” A major objective of the present study was to determine if SBV and SHUV can reassort their genome segments and generate a viable reassortant virus. To this end, plasmids encoding the genome segments of SBV were exchanged with plasmids encoding corresponding genome segments of SHUV.” These two sentences should be moved into introduction.
Line 280. “However, despite several attempts,”- should be deleted from the results. The data about how many attempts were done related to materials and methods. Please, make changes
Lines 283-292. All this paragraph have to be moved to discussion.
Lines 293-295. The sentence related to discussion.
Lines 297-298. Already written in materials and methods. Please, delete it.
Lines 299-230. I recommend to use the phrase” specifically recognizes” in this sentence.
Lines 301-302. “The successful rescue of a virus containing SBV L and S and SHUV M, suggests that such a reassortant could also emerge in the field”. The sentence should be transferred into discussion.
Line 328. Please, insert explanation for wild-life and reference for it.
Lines 328-329, “recent association with neurological disease in humans”. It is not proven. It is also not related to results. Please, change the text or insert the reference and description.
Lines 361-362. Evaluation of all simbuviruses is not right in this context. I recommend compare these two specific viruses, specifically their aa sequences of M segments. That time SBV was unknown.
Author Response
Reviewer 1
We thank the reviewer for his/her positive evaluation of our manuscript and the suggestions for improvement. Our responses to each of the comments and suggestions are given below. Line numbers in responses refer to the revised manuscript.
Comments and Suggestions for Authors
In the manuscript authors describe reverse-genetics system for reassortant SBV and SHUV. I recommend to work on structure of the manuscript all over the text.
Lines 244-260. All paragraph was to be rewritten in shorter sentences. Not clear. It should include only result without discussing the results.
All over the text not only manufacturers should be shown, but also their addresses.
Response: Thank you for your comments. The structure of the manuscript has been modified and textual changes are made as requested. The addresses of manufacturers have been added throughout the manuscript.
Abstract.
Lines 23-24. “The zoonotic potential of SHUV was recently underscored by its association with
neurological disease in humans”. It is controversial. ” I recommend to transfer the sentence into introduction or even discussion and to change the word “underscored to “unknown” or “not clear”. I suppose authors mean “febrile symptoms and neurological disease in overlooked patients in Gauteng hospitals has led to the identification of the several cases of the unrelated West Nile virus infection (Zaayman, 2012)”. If so, all these facts should be better described.
Response: Recent studies performed in South Africa have found an association between SHUV and human cases of neurological disease. The paper describing these findings is currently under review by the journal Emerging Infectious Diseases. Apart from this submitted publication, the isolation of the virus from a febrile child and detection of antibodies in veterinarians are clear indications of zoonotic potential. The manuscript of Zaayman, 2012 is not relevant for our manuscript.
Line 27. “tropism” . Not clear what authors mean. Tropism for different hosts of tissues, or both.
Response: This has been clarified in the abstract. See line 19.
Introduction.
Description of “R&D Blueprint” and consecutive description of order Bunyavirales (family, genus, serogroup) should be written separately. I recommend to pay more attention to viruses belonged to Simbu serogroup viruses, than to other viruses belonged to order Bunyavirales.
Probably, authors wanted to show SHUV as a potential human pathogen. If so, the authors should be consequent showing this data and writing their supposition.
Response: We have adapted the introduction as requested.
Line 33-34. Explanation for “R&D Blueprint “has to be added.
Response: See the response above. This part of the introduction has been removed.
Line 42-43. “The most widespread orthobunyavirus, named Oropouche virus (OROV), a member of the Simbu serogroup, causes a mild, self-limiting febrile illness [2, 3]”. Species OROV orthobunyavirus, which includes several different viruses, spreads in South America only, which is really big), causing disease in human. Describing OROV in such context is not related to the issue unless authors want to provide an example of simbuvirus causing illness in human. The most widespread virus is Akabane virus, which present in Africa, Middle East, Asia and Australia. Please, make changes.
Response: We agree with the reviewer and we have modified the text as requested.
Line 59. After the sentence “In 1977, SHUV was isolated from the brains of two horses that had succumbed to neurological disease.” reference has to be added. Please, add information in which geographic area SHUV was isolated in 1977.
Response: The appropriate references have been added as well as the geographical information. See line 66.
Line 63. I suggest to change slightly the text . “where it was associated with congenital malformations” to add “in domestic ruminants”
Response: We have added this to line 70.
Line 64. Neurological sings and fatalities were observed in young cattle only. Please, make changes.
Response: We thank the reviewer for pointing this out and have corrected this. See line 70.
Lines 84-85. “ Therefore, M segment reassortants may be capable of replicating in animals or humans that were infected previously by another member of the same serogroup [25]. “ Reference used in the text based on 24 Simbu serogroup members serological analyses. The data is incomplete. Now 32 members are already known. Please, rewrite the text.
Response: Although additional serogroups have been recognized after this paper was published, the reference is appropriate in line 81 of the revised manuscript, where we note that there is little cross-neutralization among these viruses (we do not mention the number of serogroups here). However, we totally agree with the reviewer that the same reference is not appropriate at the position that the reviewer noticed. We have removed the reference here.
Materials and Methods.
Line 111. “ACK (Ammonium-Chloride-Potassium)”- abbreviation and full name of the lysis buffer should exchange their places.
Response: We agree with the reviewer and have adjusted this as suggested. See line 109.
Line 13. RT, please first write full name and afterword in brackets abbreviation.
Response: We agree with the reviewer and have adjusted this. We do not use room temperature elsewhere in the manuscript, so we have not abbreviated it. See lines 110-111.
Line 120. Please, provide an official name of High Five cells. “Express Five medium”. Please, add manufacturer of medium.
Response: Please note that High Five™ Cells is the official name of this cell line. The cell line is derived from the parental Trichoplusia ni cell line (cabbage looper ovary) and is commonly used for the expression of recombinant proteins using the Baculovirus Expression Vector System. As is noted in lines 95-96 ‘ Culture media and supplements were obtained from Gibco (Thermo Fischer Scientific) unless indicated otherwise and the manufacturer of the medium is Gibco which belongs to Thermo Fischer scientific.
Line 148. Please, add temperature of incubation.
Response: We agree with the reviewer and have added this. See line 148.
Line 152. RT-PCR- add full meaning of the abbreviation.
Response: RT has been written in full. See line 167.
Lines 185-188. Number of permit for using animals should be added.
Response: Immunization of the rabbits was performed by the GenScript Corporation. We therefore cannot provide a permit number.
Line 198. Full name of DAPI should be added
Response: We agree with the reviewer and have added this. See line 206.
Line 222. (DAKO). Not only manufacturer should be shown, but also its address.
Response: We agree with the reviewer and have added both the mother company and address. See line 218.
Results.
Lines 227-230. “The most widely employed method to rescue (ortho)bunyaviruses from cDNA makes use of three plasmids encoding antigenomic-sense RNA of each of the three genome segments, flanked by a T7 promoter at the 5’ end and a T7 terminator and hepatitis delta virus ribozyme at the 3’ end (Fig. 1A) [29, 33, 34]. “ This sentence is not related to results.
Response: We included this so that the reader could understand the method without the need to go back to the M&M section. However, we appreciate this comment and we have removed this text.
Line 230. “These three plasmids are transfected into cells expressing T7 polymerase.” Names of plasmids have to be written. The word ”Here” has no need.
Response: This sentence as well as the word ‘here’ have been removed from the manuscript. We have added the names of the plasmids as requested. See lines 238-239.
Line 257. “Similar results were obtained” has no need to be written.
Response: Removed as requested.
Lines-276-279.” A major objective of the present study was to determine if SBV and SHUV can reassort their genome segments and generate a viable reassortant virus. To this end, plasmids encoding the genome segments of SBV were exchanged with plasmids encoding corresponding genome segments of SHUV.” These two sentences should be moved into introduction.
Response: Removed as requested. We would like to maintain one introducing sentence, since this greatly facilitates interpretation of the results (lines 280-283).
Line 280. “However, despite several attempts,”- should be deleted from the results. The data about how many attempts were done related to materials and methods. Please, make changes
Response: We agree with the reviewer and have removed this as suggested. See line 284.
Lines 283-292. All this paragraph have to be moved to discussion.
Response: We agree with the reviewer and have removed this section from the results section.
Lines 293-295. The sentence related to discussion.
Response: We respectfully disagree with the reviewer. In these sentences we discuss the results of the experiments performed in this paper. However, this paragraph was not clearly described, which was solved by restructuring this part of the manuscript according to previous comments.
Lines 297-298. Already written in materials and methods. Please, delete it.
Response: We agree with the reviewer and have adjusted the text. See lines 287-288.
Lines 299-230. I recommend to use the phrase” specifically recognizes” in this sentence.
Response: We agree with the reviewer. See line 288.
Lines 301-302. “The successful rescue of a virus containing SBV L and S and SHUV M, suggests that such a reassortant could also emerge in the field”. The sentence should be transferred into discussion.
Response: We agree with the reviewer and have removed the sentence from the results section.
Discussion.
Line 328. Please, insert explanation for wild-life and reference for it.
Response: The relevant references have been added (line 323).
Lines 328-329, “recent association with neurological disease in humans”. It is not proven. It is also not related to results. Please, change the text or insert the reference and description.
Response: Recent studies performed in South Africa have found an association between SHUV and human cases of encephalitis. We have added the reference (line 323).
Lines 361-362. Evaluation of all simbuviruses is not right in this context. I recommend compare these two specific viruses, specifically their aa sequences of M segments. That time SBV was unknown.
Response: Although we appreciate the comment of the reviewer, the sentence about limited cross-neutralization refers to Simbu serogroup viruses, not to SBV as this is what we evaluate in our manuscript. We therefore prefer to keep this sentence.
We thank this reviewer again for his/her evaluation of our manuscript and we believe the modifications have improved our manuscript considerably.
Reviewer 2 Report
The author presents an excellent tool to assess the reassortment phenomenon. However, I would like to expect some growth curve with KC cells or other culicoides or invertebrate cell lines, as well as horse or cow cell lines. Also, cross-protection or some experimental infection with mice to validate the tropism can be very useful, as suggests by authors in the abstract, introduction, and many parts of the discussion section. I think authors can use this powerful tool to address these essential questions.
Also, the Introduction and Discussion section can be reduced and focus on the primary purpose of this study.
Minor suggestion
Line 24:25: Please, rewritten and clarify this sentence for SHUV. Because NSs protein from orhobunyavirus has been shown that is an antagonist of interferon (e.g., Bunywamera, Oropouche, Schmallenberg, etc.).
Line 33:40: I suggest you remove this paragraph. I think that the introduction can be focus only on orthobunyavirus.
Line 41: Please cite Hughes et al., J Gen Virol. 2020 Jan;101(1):1-2. and Abudurexiti et al., Arch Virol 164, 1949–1965 (2019).
Line 56: If antibodies against SBV were detected in humans, I think that SBV is infectious but probably non-pathogenic. Please, redo this sentence.
Line 62: Same comment above. Infectious or pathogenic?
Line 68:76: I suggest you move to the first paragraph.
Line 77: Why you wrote “ortho)bunyaviruses” this form?
Author Response
We thank the reviewer for his/her positive evaluation of our manuscript and the suggestions for improvement. Our responses to each of the comments and suggestions are given below. Line numbers in our responses refer to those of the revised manuscript.
Comments and Suggestions for Authors
The author presents an excellent tool to assess the reassortment phenomenon. However, I would like to expect some growth curve with KC cells or other culicoides or invertebrate cell lines, as well as horse or cow cell lines. Also, cross-protection or some experimental infection with mice to validate the tropism can be very useful, as suggests by authors in the abstract, introduction, and many parts of the discussion section. I think authors can use this powerful tool to address these essential questions.
Also, the Introduction and Discussion section can be reduced and focus on the primary purpose of this study.
Response: We thank the reviewer for his/her positive evaluation of our manuscript. We agree that it would be interesting to study the growth of our reassortment virus in culicoides, horse and cow cells and to study tropism in mouse models. However, due to the limited lab access as a result of the SARS-CoV-2 outbreak we are not able to perform these experiments in the coming months. We have adjusted the introduction and discussion section also in response to Reviewer 1.
Minor suggestion
Line 24:25: Please, rewritten and clarify this sentence for SHUV. Because NSs protein from orhobunyavirus has been shown that is an antagonist of interferon (e.g., Bunywamera, Oropouche, Schmallenberg, etc.).
Response: We appreciate this comment and we have made this clear in line 25.
Line 33:40: I suggest you remove this paragraph. I think that the introduction can be focus only on orthobunyavirus.
Response: We have removed this paragraph as requested.
Line 41: Please cite Hughes et al., J Gen Virol. 2020 Jan;101(1):1-2. and Abudurexiti et al., Arch Virol 164, 1949–1965 (2019).
Response: We agree with the reviewer and have added the references. See line 35.
Line 56: If antibodies against SBV were detected in humans, I think that SBV is infectious but probably non-pathogenic. Please, redo this sentence.
Response: Serum samples collected from over 300 people who lived at or near a farm where SBV was detected were tested and no SBV-specific antibodies were found. This suggests that SBV is not infectious for humans. See lines 60-62.
Line 62: Same comment above. Infectious or pathogenic?
Response: There is much evidence that SHUV is infectious to humans, whereas its pathogenic potential is less clear. Therefore we prefer to use “infectious” here.
Line 68:76: I suggest you move to the first paragraph.
Response: We agree with the reviewer and have moved this paragraph to the beginning of the introduction. See lines 44-52.
Line 77: Why you wrote “ortho)bunyaviruses” this form?
Response: It is written like that because the ability to reassort is not only an important feature of orthobunyaviruses but of bunyaviruses in general. However as the introduction has been adapted to only discuss orthobunyaviruses as was suggested by the reviewer, it has become acceptable to just mention the reassortment as an important feature of orthobunyaviruses. Line 75.
We thank this reviewer again for his/her evaluation of our manuscript and we believe the modifications have improved our manuscript considerably.